# Understanding Vaccine Hesitancy: Insights and Improvement Strategies Drawn from a Multi-Study Review

**DOI:** 10.3390/vaccines13101003

**Published:** 2025-09-25

**Authors:** Kaitlin (Quirk) Brumbaugh, Frances Gellert, Ali H. Mokdad

**Affiliations:** 1Population Health Initiative, University of Washington, Seattle, WA 98195, USA; kquirk@uw.edu (K.B.);; 2Department of Health Metrics Sciences, University of Washington, Seattle, WA 98195, USA

**Keywords:** vaccine hesitancy, vaccine safety, public health interventions, multi-study review

## Abstract

Vaccines are among the most effective public health interventions, significantly reducing morbidity and mortality from infectious diseases. Despite their proven efficacy, vaccine hesitancy has emerged as a pressing global challenge. This review examines the drivers, barriers, and interventions associated with vaccine hesitancy and uptake, focusing on childhood vaccinations and the role of parents as primary decision-makers. Misinformation, safety concerns, and political decisions have contributed to declining vaccination rates, posing threats to public health. The article proposes targeted programs and policies to rebuild vaccine confidence, emphasizing the role of trusted messengers, health literacy, and structural reforms to reduce barriers. Recommendations highlight the importance of accurate information, open communication, and advocacy for school vaccine mandates. The conclusion stresses the urgent need to implement robust policies and community-based initiatives to ensure widespread immunization and safeguard population health.

## 1. Background

### 1.1. Overview of the Topic

Vaccination is one of the most transformative public health advancements, saving millions of lives annually by preventing the spread of infectious diseases such as measles, polio, and influenza [1]. Vaccines targeting childhood diseases, adolescent health concerns like the human papillomavirus (HPV), and other long-standing and rising threats such as influenza and COVID-19 have been pivotal in protecting communities and improving global health outcomes. Decades of evidence show that vaccines are safe, effective, and the most reliable means of protecting individuals and their communities from serious vaccine-preventable diseases [2,3].

In addition to extensive research demonstrating vaccine safety and efficacy, government agencies regulate vaccines to ensure their safety. In the United States, the Federal Drug Administration (FDA) and the U.S. Centers for Disease Control and Prevention (CDC) oversee vaccine development, requiring years of testing, issuing guidance, and continuously monitoring approved vaccines [4,5]. Despite clear evidence of their benefits, vaccine hesitancy—a reluctance or refusal to vaccinate despite the availability of vaccines—continues to rise globally, threatening these achievements and posing significant challenges to public health [6].

The World Health Organization (WHO) recognizes vaccine hesitancy as one of the top threats to global health [6], warning that it undermines herd immunity and equitable access to preventive care [7]. Vaccine hesitancy is not a monolithic challenge; it varies across populations, vaccine types, and geographic regions, influenced by sociocultural, political, and historical factors, which adds to the complexity of addressing it. Influences and drivers of vaccine hesitancy are well documented, including issues rooted in misinformation and lack of knowledge, cultural and religious beliefs, distrust in government and pharmaceutical companies, perceived risk of vaccination, personal and familial experiences, socioeconomic factors, political ideology, and social influence and peer pressure [8,9,10,11].

Hesitancy is particularly concerning among parents and healthcare workers (HCWs). Parents serve as primary decision-makers for childhood vaccinations. Their reluctance to vaccinate children increases vulnerability to outbreaks of vaccine-preventable diseases, jeopardizing not only individual health but community immunity. Many attribute the increase in vaccine hesitancy among parents to the significant spread of misinformation and disinformation surrounding vaccine safety and effectiveness, contributing to a decline in vaccine uptake regardless of the vaccine [12,13,14].

HCWs are a critical component of parental decision-making, providing education and answering parental questions on vaccination safety and efficacy. When HCWs are confident in their knowledge of vaccine safety and efficacy, they are more likely to recommend, educate, and address parental concerns [15,16,17,18]. Although vaccine hesitancy existed among healthcare workers prior to the pandemic, recent research suggests that mistrust, misinformation, and hesitancy have increased among this critical population. Factors such as education level, job position, and knowledge about vaccines influence their decision to receive a vaccination [15,16,17,18]. Despite being on the frontlines of the pandemic response, HCWs demonstrated a surprising level of vaccine hesitancy, and vaccination rates varied across the U.S. [15,19].

### 1.2. Importance of Vaccinating Children

Childhood vaccinations are crucial for preventing diseases such as measles, mumps, rubella, diphtheria, tetanus, and pertussis (whooping cough). These vaccines have a long history of safety and efficacy, supported by decades of scientific evidence. This evidence contradicts anti-vaccine rhetoric, confirming the safety of vaccine ingredients such as Thimerosal and aluminum [2,20,21,22,23,24], and the safety of Measles-containing vaccines such as the MMR [25,26,27]. Furthermore, the literature continuously shows that vaccines have no associations with autism, developmental delays, infertility, or diabetes [2,25,28,29,30,31,32,33,34,35].

Routine childhood vaccination has drastically reduced the prevalence of vaccine-preventable diseases. These vaccines target 14 different diseases and have significantly lowered the incidence of all targeted illnesses [36]. According to a 2019 study, vaccination efforts were predicted to have averted over 24 million cases of vaccine-preventable disease [36].

However, recent declines in vaccine coverage across the U.S. and the globe have already led to outbreaks, threatening decades of progress. Between 1980 and 2023, global childhood vaccination coverage saw significant increases, resulting in a 75% decline in the number of children without any vaccinations [37,38]. Between 2010 and 2019, measles vaccination rates declined in more than 100 countries [38]. Additionally, 21 out of 36 high-income countries also experienced a drop in coverage for routine childhood vaccinations [38]. Major disruptions in immunization programs during the COVID-19 pandemic, along with a decade of stagnation in vaccination coverage, have increased the risk of outbreaks [38]. Globally, more than 103 countries have experienced measles outbreaks as global vaccination rates remain lower than the necessary 95% coverage to prevent unnecessary outbreaks and deaths [38].

In the U.S., as of 29 July 2025, measles cases are nearly five times higher than the previous year, with 1300 confirmed cases. Many states in the U.S. are experiencing outbreaks, like Texas, which has been in the news for having its largest measles outbreak in 30 years [37]. Washington has seen significantly lower childhood vaccination uptake and increased exemption rates [39]. As of 25 June 2025, there have been 10 reported measles cases in the state [40]. Misinformation campaigns continue to sow fear among parents. For example, hesitancy around HPV, influenza, and MMR being linked to infertility issues and autism—a claim thoroughly debunked—has been connected to disease outbreaks among Somali, Polish, and other migrant communities [41]. Additionally, misleading and false information around COVID-19 vaccines being connected to infertility, storing tracking devices, or being used to harm and weaken communities intentionally has gained traction on social media [8].

The U.S. northern neighbor, Canada, has also seen a decrease in vaccination coverage in recent years. As of 2023, vaccination rates for MMR were 92% for the first dose and 79% for the second dose, lower than the 95% needed for herd immunity [42]. Additionally, like the U.S. and other countries, measles cases are emerging throughout the country, with a sizable outbreak in Ontario, specifically [42,43]. While studies have found that, in general, Canadians are still reporting a high level of vaccine confidence, the state of vaccine hesitancy is vulnerable and evolving [44].

### 1.3. Impact and Mishandling of the COVID-19 Pandemic

The COVID-19 pandemic played a pivotal role in the rise in vaccine hesitancy both globally and within the United States. A deeply politicized emergency response fueled widespread misinformation, eroded trust in public health and medical institutions, and further exposed longstanding systemic inequities. While the crisis demanded that leaders make rapid decisions under extreme pressure and uncertainty, the efforts under both U.S. federal administrations resulted in costly mistakes [45]. Inconsistent messaging, unclear rationale behind policy shifts, delayed data transparency, and the premature abandonment of mitigation strategies all contributed to widespread confusion and skepticism [45].

Conflicting public health communication about COVID-19 vaccine safety and efficacy seeded doubt and amplified vaccine hesitancy, allowing for anti-vaccine narratives to thrive, especially on social media [45]. Messaging often overstated the vaccines’ ability to block transmission rather than clearly explaining their primary role in preventing severe disease [45]. This communication, paired with polarizing policy choices, helped foster a more profound mistrust not only in the vaccines themselves but also in the broader healthcare system, government agencies, and science as a whole.

The CDC’s communication missteps and the broader politicization of vaccines have contributed to a crisis of credibility that remains unresolved [45]. Efforts to rebuild public trust must begin by prioritizing science over politics [45]. This includes reinforcing the independence of public health agencies such as the CDC and NIH in their decision-making processes. Additionally, we must center our goals to protect America’s health through ensuring intellectual freedom, making data public once collected, improving surveillance and data reporting, reinvigorating transmission blocking vaccine research, explaining the rationale behind policy recommendations, allowing for policy updates as we learn new science rather than waiting for it to be published, learning from other counties, empowering local health departments, empowering the CDC to engage with leading experts, and increasing analytical capacity [45,46].

Restoring trust in science and public health will require sustained investment from health agencies, the medical community, and government leadership. Effective, transparent communication, pro-science programming, and community-driven strategies are essential to counter rising vaccine hesitancy and to rebuild the public’s confidence. The erosion of trust during the pandemic has already spilled over into declining support for childhood vaccines, HPV vaccines, influenza shots, and other routine vaccinations, creating a ripple effect that jeopardizes decades of public health progress.

Although there was an initial recovery in routine childhood vaccination in many countries following the pandemic, already underserved communities were left with deepened inequities and greater barriers to care [47]. Today, we are witnessing the ongoing consequences: declining vaccination coverage, rising exemption rates, and growing resistance to vaccine mandates. Policy decisions that expand exemptions from school vaccine requirements have only exacerbated these challenges, creating opportunities for unvaccinated children to attend school and increasing the risk of disease outbreaks [48,49]. Rebuilding vaccine confidence among parents is essential not only to protect individual children but also to ensure community health and prevent the resurgence of preventable diseases. The CDC, NIH, and other government agencies need to distance themselves from the politicization of vaccines and focus on restoring trust in both vaccines and the agencies themselves.

### 1.4. Importance of Combating Vaccine Hesitancy

Vaccine hesitancy is a complex and multifaceted issue with serious public health consequences. The consequences extend far beyond individual health: they increase the risk of outbreaks, extend the circulation of vaccine-preventable diseases, and disproportionately harm communities already facing health inequities. High vaccination rates are essential for establishing herd immunity and protecting those who cannot be vaccinated due to age or medical conditions. Racial and ethnic minorities, rural populations, and economically disadvantaged groups often experience multiple barriers to vaccination, including limited access, systemic racism, and deep-rooted mistrust in healthcare systems. These forms of exclusion and mistreatment make it challenging for underserved communities to access accurate vaccine information and timely immunization services [50,51,52].

Effectively addressing vaccine hesitancy requires a clear understanding of its underlying drivers—ranging from concerns about safety and misinformation to institutional distrust and structural barriers. The variety and severity of concerns among vaccine-hesitant and anti-vaccination individuals make employing effective strategies to combat hesitancy difficult. While mandates have effectively improved coverage in many cases, they are not shown to address the underlying issues driving vaccine hesitancy [53,54]. When individuals are informed and trust the medical community, they are more likely to engage in discussions on why it is important to vaccinate and for HCWs to address concerns or misinformation. Combating misinformation, restoring trust, and fostering health literacy are crucial to addressing the underlying drivers and building vaccine confidence in hesitant communities.

Combatting hesitancy is not only vital for protecting vulnerable populations but also for rebuilding public trust and ensuring equitable access to vaccination. It demands learning from the COVID-19 pandemic, other countries, and leaning on regionally, tailored, adaptable, multi-faceted, community-informed strategies that involve trusted messengers, culturally responsive outreach, clear and consistent messaging, and long-term structural reforms to improve healthcare access and responsiveness [55].

## 2. Methods

### 2.1. Overview of Work

This review focuses on four studies conducted by the University of Washington (UW) Population Health Initiative (PHI). Synthesizing the body of work that was informed by comprehensive literature reviews, examining and expanding on common barriers to vaccination, underlying drivers of vaccine hesitancy, and effective interventions to improve vaccine uptake and equity. Each of the four studies had distinct objectives, all aimed to strengthen vaccine confidence, close equity gaps, and inform policy and program design. Table 1 summarizes the titles and aims of these studies.

The studies were built upon previous work and employed comparable methodological frameworks, using explanatory mixed-methods designs that combined an extensive literature review with quantitative and qualitative approaches. Quantitative analyses assessed vaccine performance and guided the selection of states and/or counties for qualitative inquiry, including interviews and large-scale surveys. This integrated approach provided deeper insight into systemic drivers of vaccine hesitancy, barriers to uptake, and strategies for improving coverage. Data sources included publicly available national surveys—such as the U.S. National Immunization Survey Child (NIS) [60], the Canadian Childhood National Immunization Coverage Survey (CNICS) [61], the Behavioral Risk Factor Surveillance System (BRFSS) [62]—as well as state- and county-level surveys and registries. These were used to compare immunization rates across time, regions, vaccines, and populations.

Study periods spanned five to fifteen years between 2008 and 2021. Most focused on vaccination trends in the U.S., evaluating differences across states and counties with varying policies and performance. One study incorporated Canadian data to compare programs across the two countries. Vaccines examined included childhood immunizations (e.g., MMR, DTaP), adolescent vaccines (e.g., HPV), influenza, and COVID-19. Across all studies, analyses emphasized populations with historically lower vaccine uptake or higher levels of hesitancy, including:

Racial and ethnic minority groups, underserved communities, and rural populations to assess disparities in access and trust [56];

Childhood immunization programs in both Canada and the U.S., with attention to racial inequities in vaccine delivery and outcomes [57];

Nursing staff in long-term care facilities, to explore attitudes toward COVID-19 vaccination [58];

Parents of children and adolescents, particularly in politically and geographically diverse regions (Florida and New York), to examine HPV attitudes and behaviors [59].

In-depth qualitative inquiry—including key informant interviews, focus groups, and large-scale surveys—examined barriers to vaccination, drivers of hesitancy, and strategies to improve uptake across regions and populations with differing vaccine performance. Interviews with public health officials, physicians, and community health advocates highlighted both persistent challenges and effective approaches to strengthening confidence and coverage. Key themes were identified through consensus-building among researchers, development of a deductive-inductive codebook, and a systematic review of coded transcripts.

All studies were funded by research grants from the Investigator-Initiated Studies Program of Merck Sharp & Dohme Corp.

### 2.2. Identifying Common Themes

This review synthesizes findings from four studies informed by a comprehensive literature review and presents recommendations for addressing the complex factors that shape vaccine hesitancy and uptake.

To identify shared themes, we conducted a comparative thematic synthesis. Analytic outputs—including coded transcripts, survey findings, and quantitative results—were jointly reviewed to assess overlapping patterns. Through consensus-building discussions, we reconciled study-specific findings and distilled key drivers and barriers that consistently emerged across settings and vaccine types.

The findings summarized in this review represent the themes most frequently observed across studies and most commonly reported by respondents.

## 3. Findings from Multi-Study Review

Although each of the four studies differed in terms of population, setting, and vaccine type, several consistent themes emerged. Together, these studies highlight the importance of localized, data-driven approaches in addressing vaccine hesitancy, data limitations and inequities, and barriers to vaccination. Drawing on existing literature and combined evidence, this review provides a comprehensive understanding of the issues and outlines strategies and recommendations for addressing vaccine hesitancy, reducing disparities, and increasing vaccine confidence and uptake across diverse communities.

### 3.1. Drivers of Vaccine Hesitancy

Across studies and populations, several key drivers of vaccine hesitancy emerged: safety concerns, efficacy doubts, misinformation and social media, distrust in institutions, religious and cultural beliefs, and agency and autonomy. These drivers reflect the complex interplay of individual beliefs, societal influences, and systemic factors.

Many individuals expressed fear of side effects, doubt about the safety of vaccines, or anecdotal experiences of adverse reactions. For example, concerns about rare but widely publicized side effects of the HPV and COVID-19 vaccines fueled hesitancy. Misinformation amplified these fears, with social media platforms spreading false claims about vaccine ingredients, infertility, and long-term effects.

Efficacy doubts also contributed to hesitancy, with some parents questioning whether vaccines effectively prevent disease or are necessary, given a perception of the declining prevalence of certain illnesses. Misunderstandings about vaccine mechanisms—such as their ability to reduce disease severity rather than guarantee immunity—fueled skepticism, especially for COVID-19 and influenza vaccines.

Historical and systemic racism, coercive policies, and unethical medical practices—such as the Tuskegee Syphilis Study—fueled distrust in government and healthcare systems, particularly among affected communities. This distrust extended to vaccine campaigns perceived as disconnected from community needs or prioritizing profits over public health.

Religious and cultural beliefs shaped vaccine hesitancy in some populations. Certain religious groups resisted vaccines due to concerns about their compatibility with spiritual beliefs or alternative approaches to medicine. Cultural norms also influenced perceptions of vaccines, including stigma around HPV vaccination for adolescents.

Many individuals reported feeling pressured or coerced into vaccination, undermining their ability to make informed decisions. Changing vaccine mandates, lack of time to discuss concerns with healthcare providers, and rushed vaccination appointments further undermined trust and contributed to hesitancy.

### 3.2. Data Limitations and Inequities in Vaccination

Across our research, vaccine registry data and population-based surveys consistently underscored the importance of analyzing vaccination coverage at the local level and disaggregating data by population subgroup. Granular data is essential for identifying inequities, tailoring interventions, and monitoring progress. However, our review also revealed critical limitations in data systems that hinder efforts to understand and address vaccine disparities fully.

Several studies identified gaps in the quality and completeness of vaccination data. Many registries lack standardized reporting protocols, and key demographic indicators—such as race, ethnicity, income, language, and geographic location—are often missing or inconsistently collected. These deficiencies limit the ability of public health systems to detect and respond to disparities in vaccine access or uptake. Technical and infrastructural barriers, particularly in rural and under-resourced settings, further compound these challenges. Fragmented health systems and limited interoperability between electronic health records and immunization registries result in delays, data loss, or incomplete reporting.

In studies focused on racial and ethnic disparities, we found data security concerns and the principles of Indigenous data sovereignty significantly influence what data is collected, how it is shared, and with whom. Protecting community-controlled data is essential to ethical research and policymaking. At the same time, it highlights the importance of engaging communities in both the planning and implementation of vaccination programs. These efforts must be grounded in trust, responsive to community-identified needs, and guided by transparent and reciprocal data-sharing practices. Our findings reaffirm that ethical data use must go hand-in-hand with effective, community-informed public health planning.

Nationally, data from the National Immunization Survey—Child (NIS-Child) [60], as reported in our previous Canada-U.S. comparative study [56], show that childhood vaccination coverage for children aged 19 months to 3 years remained relatively stable between 2011 and 2021 (Figure 1). In 2011, vaccination rates for DTP, MMR, and Hep B were 86%, 92%, and 91%, respectively; in 2021, these values were 86%, 94%, and 94%, respectively.

However, trends diverged when disaggregated by race/ethnicity and socioeconomic status. Figure 2, drawn from our study on racial and socioeconomic disparities in U.S. childhood vaccinations [59], presents the percentage difference in coverage across racial/ethnic and income groups, relative to White children of the same income level. A value of zero indicates no difference; a negative value reflects a greater decline in coverage relative to White children; and a positive value reflects a greater improvement.

Our analysis shows that high-income Black children experienced more notable improvements in vaccine coverage than their low-income counterparts, while Hispanic children showed less improvement overall. Racial disparities were especially pronounced in MMR coverage among medium- and low-income Black children, and in DTP coverage among medium- and low-income Hispanic children. These patterns reflect the compounded effects of structural inequities that influence vaccine access and health outcomes.

A county-level analysis of HPV vaccine hesitancy in New York and Florida further illustrated the influence of sociodemographic factors—including gender, age, race, religion, income, and employment—on childhood and adolescent vaccination rates [58]. Female and younger parents (aged 18–34) were more likely to report that children aged 13 and over had received at least one HPV vaccine dose. White respondents were more likely than Black or Indigenous respondents to have vaccinated children in this age group. Compared to Christian respondents, atheist/agnostic and Muslim respondents were more likely to have vaccinated children aged 11 and older. Higher-income respondents (earning over $75,000) and those who were employed were more likely than lower-income (<$35,000) or unemployed respondents to report vaccinated children. These patterns underscore the importance of nuanced, local-level approaches to vaccine equity.

Despite data limitations, the 2021 Childhood National Immunization Coverage Survey (CNICS) [61] shows that 2-year-old childhood vaccination rates in Canada also vary by race and ethnicity (Figure 3) [56]. For example, Black children had lower DTP coverage (66%) compared to White children (78%), while Chinese children had higher coverage (87%). MMR coverage among Filipino children was also higher (96%) than among White children (92%). Although additional differences in vaccine uptake were observed across population groups, many were not statistically significant due to small sample sizes.

While our findings highlight persistent disparities in vaccine coverage, they also reveal the constraints posed by limited early childhood data and a lack of disaggregated race and ethnicity data at the local level. These limitations hinder the ability to generate a full picture of inequities in vaccination. To close existing coverage gaps, improved data quality, consistent demographic reporting, and robust monitoring systems are urgently needed—particularly in communities that have been historically underserved and continue to fall behind.

### 3.3. Barriers to Vaccination

Barriers to vaccine uptake were multifaceted and often intertwined with the drivers of hesitancy. Common barriers to vaccination across the studies included social determinants of health, structural barriers, data limitations, policy constraints, and messaging and education. Table 2 describes the commonalities in findings and provides a description of key themes, sub-themes, and their corresponding descriptions.

Particularly for underserved communities, access challenges compounded by social determinants of health were significant. Financial constraints, lack of transportation, and competing priorities such as work schedules or childcare responsibilities frequently made it difficult for parents to schedule vaccination appointments.

These challenges were rooted in structural barriers, including systemic racism and fragmented healthcare systems, limiting the availability of vaccination opportunities in rural and economically disadvantaged areas. Healthcare workforce limitations also hindered vaccine delivery, with a shortage of primary care providers restricting access and inconsistent communication from healthcare workers undermining parental confidence. Providers often lacked the time or training to address hesitancy during appointments, leaving patients’ concerns unaddressed.

Inadequate data collection further complicated vaccination efforts, with limited disaggregation by race, ethnicity, and other variables reducing the ability to target interventions effectively. Sensitivity around Indigenous data in Canada posed additional challenges, highlighting the need for culturally competent approaches to vaccination outreach.

Policy has a significant influence on vaccination uptake. Policies relating to healthcare, education, and public health can either facilitate or hinder access to vaccinations. Policy barriers can be particularly pronounced in rural or already underserved areas. Eligibility criteria may exclude vulnerable populations, such as low-income families or undocumented immigrants, who may hesitate to seek healthcare services.

Funding and resource allocation also affect vaccination rates; inadequate funding for public health initiatives can lead to fewer outreach and educational campaigns, perpetuating misinformation and hesitancy. The political climate can further complicate vaccine acceptance. Polarized views on public health may diminish trust in vaccine safety, primarily when policymakers do not adequately address misinformation.

Furthermore, school-based policies play a significant role in routine childhood vaccination uptake. Each state has slightly different requirements, mandates, allowable exemptions, and processes for receiving exemptions. These policies can result in clusters of unvaccinated individuals, heightening the risk of outbreaks and undermining herd immunity.

Overcoming these policy barriers demands collaboration among health officials, community leaders, and policymakers to create inclusive and informed vaccination policies that enhance uptake and protect public health.

Regardless of vaccine or target audience, our studies identified that educational interventions are incorporated at all levels of intervention and valued by many providers as a key intervention. Much of public health messaging is focused on education. Education focuses on countering misinformation, improving health literacy, fostering agency, and providing culturally relevant and accessible resources.

While emphasizing the importance of messaging and education, studies identified a lack of knowledge among decision-makers and healthcare providers, particularly regarding newer vaccines such as HPV and COVID-19. Parents felt there was inadequate time to discuss concerns with healthcare providers or felt coerced into vaccination.

### 3.4. Interventions to Improve Uptake

Through our review, we identified a range of interventions to address vaccine hesitancy and increase uptake, consistently emphasizing the need for strategies tailored to the unique needs of specific populations. Across diverse contexts, several standard approaches emerged. These included the implementation of culturally and linguistically responsive educational campaigns, the use of trusted messengers, and strong provider recommendations. Other key strategies involved improving workforce diversity and representation, engaging communities in vaccine planning and delivery, reducing structural barriers to access, strengthening data systems, and promoting equitable policy changes.

Community engagement emerged as a cornerstone of effective interventions, with collaborations between healthcare providers and trusted community leaders fostering culturally relevant messaging and outreach. For example, partnerships with faith-based organizations helped address religious concerns, while collaborations with local schools supported HPV vaccination programs.

Educational campaigns were another critical strategy, improving health literacy and empowering individuals to make informed decisions. These campaigns often employed storytelling and social media outreach to counter misinformation and build trust. Provider recommendations also proved influential, with trusted healthcare workers playing a key role in motivating individuals to vaccinate. Accessibility measures, such as mobile clinics, extended clinic hours, and on-site vaccination at community events, reduced logistical barriers and increased coverage rates in underserved areas.

Policy and structural changes were essential for addressing systemic barriers to vaccination. Efforts to improve funding for vaccination programs, create unified vaccine delivery systems, and address social determinants of health were critical steps in promoting equitable access. Data enhancement, including the collection of disaggregated data by race and ethnicity, allowed for better targeting of underserved populations and improved understanding of vaccine coverage gaps.

Together, these interventions reflect a multifaceted approach to rebuilding trust, addressing systemic barriers, and improving vaccine confidence and coverage.

## 4. Discussion: Recommendations for Programs and Policies

### 4.1. Building Trust Through Education and Communication

One of the most effective ways to combat vaccine hesitancy is through education and open communication. Parents need access to accurate information about the safety, efficacy, and necessity of childhood vaccines. Healthcare providers play a key role in delivering this information, fostering a welcoming environment for parents to ask questions and discuss concerns. Longer appointment times, culturally sensitive communication, and personalized interactions can help rebuild trust.

Educational campaigns should be tailored to address specific misconceptions, such as debunking the myth that vaccines cause autism. These campaigns can leverage storytelling and social media platforms to reach hesitant parents, counter misinformation, and promote vaccine confidence. Collaborations with schools and parent–teacher associations offer additional outreach opportunities, particularly for childhood vaccination programs.

### 4.2. Improving Accessibility to Vaccination Services

Structural barriers to vaccination must be addressed to improve uptake among hesitant parents. Expanding mobile clinics, extending clinic hours, and offering onsite vaccination at schools and community events can reduce logistical challenges. Free vaccines and transportation assistance can further alleviate financial and practical barriers for low-income families.

Healthcare systems must prioritize culturally competent approaches to vaccination outreach, ensuring that services are inclusive and accessible to marginalized populations. Addressing the shortage of primary care providers, particularly in rural areas, is critical to improving accessibility and reducing disparities in vaccine delivery.

### 4.3. Safeguarding School Vaccine Mandates

School vaccine mandates are a cornerstone of public health policy, ensuring high immunization rates and protecting children from vaccine-preventable diseases. However, political decisions expanding exemptions threaten these mandates, increasing the risk of outbreaks and eroding community immunity. Policymakers must resist efforts to weaken vaccine requirements for school entry and instead strengthen policies that promote universal vaccination.

Parents can play an active role in advocating for school vaccine mandates, emphasizing their importance for protecting children and communities. Public health agencies should work with schools to implement vaccination programs that are convenient, accessible, and well-publicized, reducing barriers for families and encouraging compliance.

### 4.4. Policy and Structural Changes

Policies should be created to address the underlying social determinants of health, improve funding for vaccination programs, and create unified vaccine delivery systems, which are essential for systemic change. Existing and new policy and vaccine legislation must be guided by science and the common good in mind. Legislation regarding vaccines should prioritize public health and the well-being of our communities, rather than serving narrow interests. Only through thoughtful and equitable policies can we hope to create a healthier future for all. Addressing these challenges with a comprehensive, science-driven approach can achieve lasting systemic changes that benefit everyone.

### 4.5. Community-Based Interventions

Community engagement is essential for addressing vaccine hesitancy among parents. Collaborating with trusted messengers, such as religious leaders, teachers, and local organizations, can foster culturally relevant messaging and outreach. For example, partnerships with faith-based organizations can address religious concerns, while collaborations with community centers can promote vaccination at local events.

Programs that empower parents to become advocates for vaccination within their communities can also be effective. Parent-led initiatives that share personal stories about the benefits of vaccination can build trust and inspire others to vaccinate their children.

### 4.6. Charting a Way Forward

Looking ahead, addressing vaccine hesitancy will require a shift from reactive, piecemeal solutions to proactive, sustained, and evidence-based action. Building trust must become an ongoing priority, with investment in transparent communication, meaningful community partnerships, and the empowerment of trusted local messengers. Enhancing health literacy through culturally relevant education, improving provider training, and ensuring open, judgment-free discussions about vaccines are critical next steps. At the same time, public health agencies and policymakers must prioritize the modernization of immunization data systems, enabling real-time identification of disparities and more precise targeting of interventions.

Equity should be at the center of all future efforts. This means not only reducing logistical and financial barriers, but also confronting structural inequities, supporting diverse workforce representation, and ensuring that marginalized voices are partners in program design and policy development. Policymakers should reinforce the importance of school vaccine mandates, resist the expansion of non-medical exemptions, and invest in systems that make vaccination easy and accessible for all. As the landscape of infectious disease threats evolves, a coordinated, science-driven strategy—grounded in local context and community engagement—will be essential for restoring confidence and safeguarding public health for generations to come.

### 4.7. Limitations

Vaccine hesitancy, unfortunately, is a moving target and changing rapidly. We reported that as of 2024, the current political situation in many countries, especially the U.S., and the impact of COVID-19, are changing the perceptions of vaccines. Therefore, we need to regularly monitor vaccine uptake and the drivers of vaccine hesitancy. Additionally, in one of our studies, we discuss a comparison between Canada and the U.S. We report the commonalities found in each study, acknowledging that the U.S. and Canada have different health and political systems that influence hesitancy and barriers. Therefore, not all the findings discussed may be generalizable to both countries.

## 5. Conclusions

Vaccine hesitancy remains one of the most pressing challenges to public health in the twenty-first century, threatening hard-won progress against vaccine-preventable diseases. As this review demonstrates, overcoming hesitancy requires more than correcting misinformation or implementing one-size-fits-all policies. It demands a comprehensive, equity-focused approach that rebuilds public trust, addresses systemic barriers, and centers community voices in the development and delivery of interventions. By investing in transparent communication, robust data infrastructure, accessible vaccination services, and culturally competent programs, we can begin to close the gaps in vaccine confidence and coverage. We need to be vigilant and have strong monitoring of coverage, attitudes, barriers, and intention to vaccinate to provide policymakers and the public with the needed information to maintain the success of vaccines. The path forward will not be simple, but with sustained commitment and collaboration across sectors, it is possible to ensure that every community can benefit from the full promise of vaccination.

## Figures and Tables

**Figure 1 vaccines-13-01003-f001:**
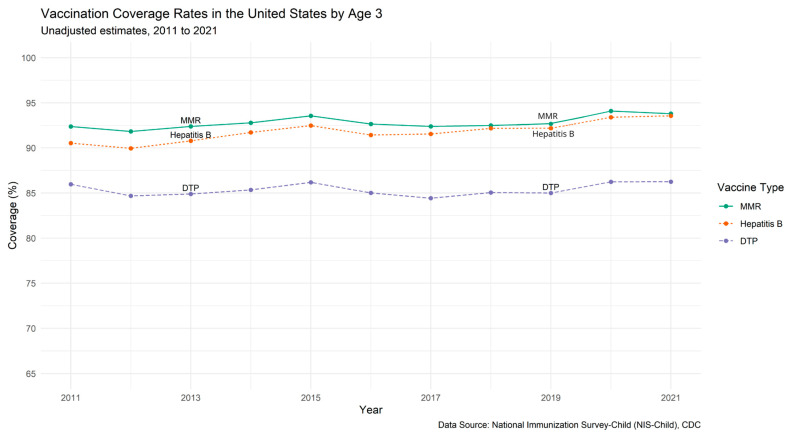
Vaccination Coverage Rates in the United States by Age 3 (NIS-Child).

**Figure 2 vaccines-13-01003-f002:**
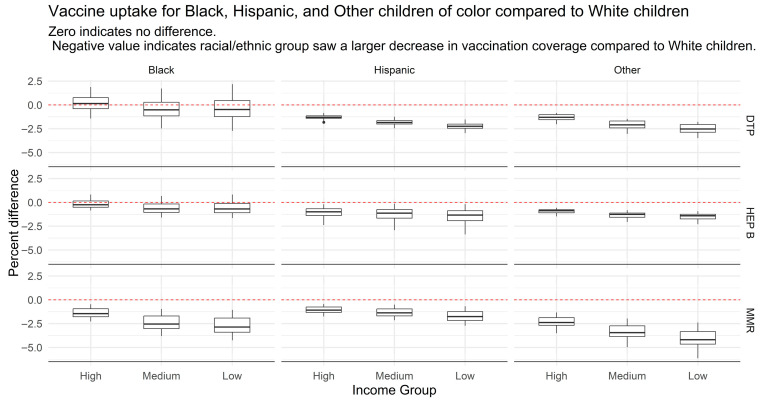
Vaccine Uptake for Black, Hispanic, and Other Children of Color Compared with White Children by Income Group (NIS-Child).

**Figure 3 vaccines-13-01003-f003:**
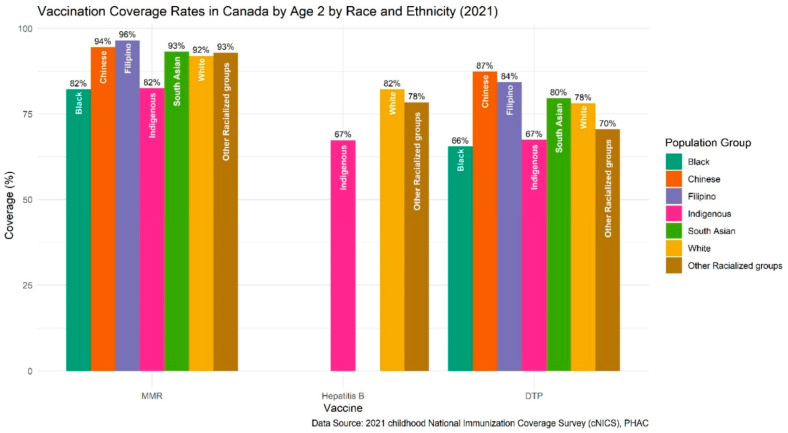
Vaccination Coverage Rates in Canada by Age 2 by Race and Ethnicity (CNICS).

**Table 1 vaccines-13-01003-t001:** Four Study Aims.

Study	Aim 1	Aim 2	Aim 3
Achieving Equity in Childhood Vaccination: A Mixed-Methods Study of Immunization Programs, Policies, and Coverage in 3 US States (2024) [56]	Measure changes in immunization equity over the last 15 years, identifying nine representative counties for qualitative analysis: three each of strong performers, weak performers, and median performers.	Identify which public health interventions or public policies have most effectively improved racial immunization equity.	Identify features of interventions that seem to provide “transferable benefits” to other vaccine-preventable disease contexts.
Bridging the gap: A mixed-methods analysis of Canadian and U.S. immunization programs for enhancing racial equity in childhood vaccinations (2025) [57]	Conduct a comprehensive review to track changes in immunization rates, policies, and practices over the last decade, with particular attention paid to rates among racial minorities in target areas, and how these compare to rates among whites.	Identify which public health interventions or public policies have most effectively improved racial immunization equity.	Compare data with findings from the United States to refine recommendations and produce outputs with broad, multinational relevance.
Drivers of HPV vaccine hesitancy in New York and Florida (2025) [58]	Measure changes in HPV immunization coverage from 2008 to 2020, disaggregating results to identify and quantify the impact of various demographic characteristics upon willingness to vaccinate. Identify six representative counties for qualitative analysis: two each of strong performers, weak performers, and median performers.	Identify which interventions or policies have most effectively improved HPV immunization rates in six representative U.S. counties.	Compare data with findings from the European Union to refine recommendations and produce outputs with broad, multinational relevance.
Identifying Emerging Drivers and Interventions to Reduce Vaccine Hesitancy Among Long-Term Care Facility Nursing Staff (2025) [59]	Measure changes in immunization rates over the last 5 years (2016–2021), identifying nine representative counties for qualitative analysis: three each of strong performers, weak performers, and median performers.	Compare the rates of vaccination among nursing staff with those of their local community in all nine sites.	Identify primary drivers of vaccine hesitancy among nursing staff, and key interventions that have most effectively contributed to improving vaccination rates for COVID-19 and/or annual influenza in the nine representative U.S. counties.

**Table 2 vaccines-13-01003-t002:** Commonalities Across Studies: Key Themes and Definitions.

Theme	Subtheme	Definition
Barriers to vaccination	Social determinants of health	Challenges rooted in social and economic conditions, including financial constraints, lack of insurance, limited transportation options, inflexible work schedules, competing demands, and language or literacy barriers that reduce access to vaccination.
	Structural barriers	Physical and systemic obstacles to vaccination access, such as limited availability of healthcare providers in rural or underserved urban areas, clinic hours that do not accommodate working individuals, brief or rushed clinical interactions.
	Data limitations	Gaps in the collection, quality, and use of data hinder the ability to identify disparities, track progress, tailor interventions, and assess the effectiveness of vaccination programs.
	Policy constraints	Restrictive or fragmented policies, underfunded and understaffed public health departments, and a complex healthcare infrastructure that together limit coordination, outreach, and the implementation of effective vaccination strategies.
	Messaging and education	Lack of clear, consistent, and culturally relevant public health communication; limited access to quality health education; and insufficient understanding of vaccine science, safety, and development processes contribute to confusion and reduced confidence in vaccines.
Drivers of vaccine hesitancy	Misinformation	False or misleading information—often spread through social media and informal networks—that promotes inaccurate claims about vaccine safety, ingredients, or effectiveness, contributing to fear and confusion.
	Concerns with safety or side effects	Fears about immediate or long-term side effects of vaccines, including beliefs that vaccines may cause chronic conditions such as autism or infertility, despite scientific evidence to the contrary.
	Perception of low vaccine efficacy or importance	Belief that vaccines are ineffective in preventing illness, or that the targeted disease is not serious or pervasive enough to warrant vaccination.
	Distrust in institutions	Deep-rooted skepticism toward healthcare systems and government agencies due to historical and ongoing experiences of racism, neglect, and marginalization.
	Impact of COVID-19 pandemic	The COVID-19 pandemic has heightened public doubt through perceived inconsistencies in public health guidance, concerns about rapid vaccine development, increased exposure to misinformation, and lasting distrust in health authorities.
	Religious, cultural, or political reasons	Vaccine hesitancy influenced by religion, cultural beliefs, or political ideology, including preferences for traditional medicine, beliefs in divine protection, or resistance tied to political identity.
	Agency and autonomy	Resistance rooted in feeling coerced, manipulated, or lacking control in the decision-making process, including concerns about mandates and the right to make personal health choices.
Interventions to improve vaccination uptake	Population-specific, tailored strategies	Design interventions that are responsive to the unique needs, concerns, and contexts of different populations, avoiding one-size-fits-all approaches and allowing for flexibility and adaptation over time.
	Community engagement	Engage communities directly in the design, planning, and implementation of vaccination initiatives. Collaborate with trusted local organizations to build relationships and increase vaccine acceptance and access.
	Improve accessibility	Expand vaccine access through mobile clinics, evening and weekend hours, school-based vaccinations, and community events. Provide support services such as transportation and childcare. Eliminate out-of-pocket costs for recommended vaccines.
	Culturally and linguistically competent	Ensure vaccine information is culturally relevant, written in plain language, and available in the languages spoken by the target populations. Clinical interactions should also reflect cultural humility and respect.
	Trusted messengers and workforce representation	Share vaccine information through trusted community voices. Build a diverse healthcare workforce on all levels that reflects the racial, cultural, and linguistic backgrounds of the communities it serves.
	Provider recommendation	Strengthen vaccine uptake through clear and confident provider recommendations, personalized interactions, motivational interviewing, and sufficient time to answer patient questions and concerns.
	Education and communication	Develop community-centered education and outreach efforts that address misinformation, improve health literacy, and connect people to vaccine services. Use storytelling, social media, and short videos to increase reach and engagement.
	Policy priorities	Implement policy measures such as school-entry vaccine requirements and other mandates that improve equity and support higher vaccination rates.
	Data quality and enhancement	Strengthen vaccination data systems by improving local-level data collection, disaggregating by social and demographic factors, and using data to guide targeted outreach and intervention strategies.

## Data Availability

Data is available at https://www.cdc.gov/brfss/annual_data/annual_2021.html, https://www.cdc.gov/nis/about/index.html, and https://www23.statcan.gc.ca/imdb/p2SV.pl?Function=getSurvey&SDDS=5185, all accessed on 19 August 2025.

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
