# Peer review of "Understanding Vaccine Hesitancy: Insights and Improvement Strategies Drawn from a Multi-Study Review"

_vaccines, 2025, doi:10.3390/vaccines13101003_

Round 1
Reviewer 1 Report
Comments and Suggestions for Authors
This manuscript tackles the highly relevant issue of vaccine hesitancy, synthesizing multiple studies. It focuses on common barriers, drivers, and potential interventions to strengthen vaccine confidence. The review is timely, clearly writen, and well-structured, with a strong emphasis on the policy implications of declining coverage. However, the manuscript would benefit from greater methodological transparency, broader inclusion of international evidence, and a sharper critical appraisal of the cited studies. My reccomendations are:
Regarding methodological clarity: section 2 describes the “explanatory sequential mixed-methods designs” but remains vague. Essential details are missing such as sample sizes and demographics of survey participants, recruitment strategies and response rates etc
Much of the manuscript is descriptive. It would be strengthened by a more explicit assessment of the relative weight of findings: Which barriers or drivers were most consistently observed across contexts? Which interventions demonstrated measurable effects, and which remain largely aspirational? A structured evaluation (e.g. indicating where evidence is strong, limited, or inconsistent) would improve the scientific rigour.
The recommendations (education, mandates, trusted messengers, structural reforms) are aligned with public health priorities but are presented in general terms. For example, Which strategies documented in the cited studies significantly improved uptake?
To enrich the global perspective, I recommend integrating recent European studies that document vaccination practices and coverage trends in Italy, such as when addressing regional and policy variations, the Signorelli et al. 2024 paper illustrates best practices from Italian regions, useful for comparison with U.S. state-level heterogeneity. Or when discuss Interventions and Recommendations
Signorelli C, Pennisi F, D’Amelio AC, Conversano M, Cinquetti S, Blandi L, Rezza G. Vaccinating in Different Settings: Best Practices from Italian Regions. Vaccines (Basel). 2024 Dec 28;13(1):16. doi:10.3390/vaccines13010016.
Pennisi F, Silenzi A, Mammone A, Siddu A, Odone A, Sabbatucci M, Orioli R, D’Amelio AC, Maraglino F, Rezza G, Signorelli C. Childhood Immunization Coverage Before, During and After the COVID-19 Pandemic in Italy. Vaccines (Basel). 2025 Jun 25;13(7):683. doi:10.3390/vaccines13070683.
Consider incorporating systematic reviews and global analyses from 2023–2025
Author Response
Comment 1: Regarding methodological clarity: section 2 describes the “explanatory sequential mixed-methods designs” but remains vague. Essential details are missing such as sample sizes and demographics of survey participants, recruitment strategies and response rates etc.
Response: Thank you for your suggestions. We have provided additional information and clarity on the studies that we have conducted in section 2.1 (lines XXX) and 3 (lines XXX). We also included a table showing the studies and their corresponding aims (Table 1) to provide further understanding on what the studies entailed.
Comment 2: Much of the manuscript is descriptive. It would be strengthened by a more explicit assessment of the relative weight of findings: Which barriers or drivers were most consistently observed across contexts? Which interventions demonstrated measurable effects, and which remain largely aspirational? A structured evaluation (e.g. indicating where evidence is strong, limited, or inconsistent) would improve the scientific rigour.
Response: The reviewer raises a very important point. In all our qualitative work, we received information on the barriers, and we focused our data presentation on the ones that were mentioned the most or brought up the most by our respondents. As you can imagine our topics have lots of potential answers/explanations and our study team focused on the main issues that were common across respondents.
Comment 3: The recommendations (education, mandates, trusted messengers, structural reforms) are aligned with public health priorities but are presented in general terms. For example, Which strategies documented in the cited studies significantly improved uptake?
Response: We did not measure the impact of recommended interventions; however, this highlights an opportunity for additional research on the topic - to evaluate how specific recommendations influence vaccination uptake in different communities. Strategies documented across the studies arose as most frequently cited interventions throughout key informant interviews with public health professionals, physicians, and academics. We have elaborated on how key themes were identified in section 2.1.
Comment 4: To enrich the global perspective, I recommend integrating recent European studies that document vaccination practices and coverage trends in Italy, such as when addressing regional and policy variations, the Signorelli et al. 2024 paper illustrates best practices from Italian regions, useful for comparison with U.S. state-level heterogeneity. Or when discussing Interventions and Recommendations.
Signorelli C, Pennisi F, D’Amelio AC, Conversano M, Cinquetti S, Blandi L, Rezza G. Vaccinating in Different Settings: Best Practices from Italian Regions. Vaccines (Basel). 2024 Dec 28;13(1):16. doi:10.3390/vaccines13010016. 
Pennisi F, Silenzi A, Mammone A, Siddu A, Odone A, Sabbatucci M, Orioli R, D’Amelio AC, Maraglino F, Rezza G, Signorelli C. Childhood Immunization Coverage Before, During and After the COVID-19 Pandemic in Italy. Vaccines (Basel). 2025 Jun 25;13(7):683. doi:10.3390/vaccines13070683.
Consider incorporating systematic reviews and global analyses from 2023–2025 
Response: We appreciate the sharing of studies to discuss and strengthen our manuscript. We have included these in the text, but we are careful to extend beyond our scope, as our studies are focused on the U.S. and Canada.
Reviewer 2 Report
Comments and Suggestions for Authors
The paper has a focus on offering insights from study review on vaccine hesitancy.
paragraph at line 39 -This is a wide-ranging topic area and this is some of the aspects of it. Not all.
around line 130 - "Restoring trust in science and public health will require sustained investment from
health agencies, the medical community, and government leadership" - In a paper with this topic, with your already describing the misinformation and politicization of COVID-19 response that resulted in nothing short of a debacle, what makes you think that 'restoring trust' is even a goal for government leadership, let alone others you mentioned? Are we just supposed to take this as given? The current US administration is quite literally demanding studies be retracted so that a vaccine conspiracy can be exposed. That would tend to undermine the seriousness of such a statement.
The point is being made because vaccine hesitancy has a growing literature and this paper basically provides a gloss on the topic, from a literature review perspective.
Section 1.4 - this is too brief a summary for this sub-topic, since the whole paper (arguably?) hinges on it.
Section 2.1 - this is state level work. Perhaps it is possible to draw a line somewhere between federal goals/views and state views? They are different, are they not? Not to mention US and global views and any differences there (likely too much to evaluate in the course of one paper).
also 2.1 - it's not clear what studies you evaluated? What was your search?
Section 3 - this is ok for a summary but there's no backup/tables/data?
Section 4 - I do no understand what method(s) were used to evaluate the studies and arrive at these commonalities. Was this a qualitative study? It's not clear how the study was designed or undertaken.
Are figures 1 and 2 pre-existing figures or are they your figures?
Section 5 - the references here seem like the review was not necessarily systematic - rather, there was some cherry-picking in terms of the studies. What was being looked for and found, versus what the studies as a group were suggesting? Again, more needs to be done in terms of clarifying the method and explaining what occurred with that.
Section 6 - these barriers - there are no references back to studies. What was the rationale with this? How did you discern the barriers from the literature?
Table 1 - how were these identified/defined as 'key'?
Section 7 - it is difficult for me to discern the quality of the discussion, given my lack of clarity on the method and presentation of analysis/findings.
I would suggest though that 'building trust' might be different in the eyes of the beholder - and how policy is written leads to strikingly different outcomes? Also, what the public actually thinks about vaccines as a result of the information/misinformation/sources of information they rely upon.
Author Response
Comment 1: paragraph at line 39 -This is a wide-ranging topic area and this is some of the aspects of it. Not all.
Response: We thank the reviewer for the comment and have included language to clarify that parental and HCW hesitancy are just aspects of vaccine hesitancy that are particularly concerning as parents are key decision makers and that HCWs are critical components to that decision making process. This discussion leads us to the next section on the importance of vaccinating children. We hope that these changes are satisfactory.
Comment 2: around line 130 - "Restoring trust in science and public health will require sustained investment from health agencies, the medical community, and government leadership" - In a paper with this topic, with your already describing the misinformation and politicization of COVID-19 response that resulted in nothing short of a debacle, what makes you think that 'restoring trust' is even a goal for government leadership, let alone others you mentioned? Are we just supposed to take this as given? The current US administration is quite literally demanding that studies be retracted so that a vaccine conspiracy can be exposed. That would tend to undermine the seriousness of such a statement.
Response: Thank you for this comment. Trust has been documented as a key issue throughout our research, that the public and parents will not adhere to public health recommendations if they do not trust the messenger. Even in times of political challenges, we need to adhere to science and our work which leads us to the recommendation of restoring trust within these various settings.
Comment 3: The point is being made because vaccine hesitancy has a growing literature, and this paper basically provides a gloss on the topic, from a literature review perspective. 
Response: Thank you for this comment. As you know hesitancy is a changing and, in a way, a moving target with COVID-19 and some of the political changes especially in the U.S. We reported on recent findings and the most recent attitude, but we do acknowledge that there are rapid changes, and we need to keep studying hesitancy. We added a sentence in our limitations to address this excellent point. We believe that we have strengthened the manuscript in clarity.
Comment 4: Section 1.4 - this is too brief a summary for this sub-topic, since the whole paper (arguably?) hinges on it.
Response: We appreciate your comment and have expanded the section accordingly.
Comment 5: Section 2.1 - this is state level work. Perhaps it is possible to draw a line somewhere between federal goals/views and state views? They are different, are they not? Not to mention US and global views and any differences there (likely too much to evaluate in the course of one paper).
Response: Thank you for this comment. We have addressed this by providing more context in the overview of our work to clarify the scope of each study. Additionally, federal money coming to states in the U.S. is marked for certain activities so in a way most of the work is done by federal-level mandates. The federal government cannot tell a state what to do but can mark funds for specific activities. Some rich states have different policies and activities that they fund but unfortunately in the U.S. starting 2020 most states relied heavily on federal funding for their health activities due to poor economy.
Comment 6: also 2.1 - it's not clear what studies you evaluated? What was your search?
Response: We evaluated four studies. These four studies were selected because of our close association as researchers who conducted the prior research at the University of Washington Population Health Initiative. We have added additional text in section 2.1 explaining the study aims, methodologies, scope of these studies to clarify this for readers.
Comment 7: Section 3 - this is ok for a summary but there's no backup/tables/data?
Response: We have addressed this comment by including a description on our process of identifying commonalities in findings through a comparative thematic analysis and synthesis. We do not have further data to include here. Table 2 identifies and defines the common themes and findings across the qualitative components in each of the four studies.
Comment 8: Section 4 - I do no understand what method(s) were used to evaluate the studies and arrive at these commonalities. Was this a qualitative study? It's not clear how the study was designed or undertaken.
Response: We appreciate the opportunity to add clarity to the manuscript. All four studies employed mixed-methods approach, quantitative exploration and qualitative inquiry. We have added more clarity to each study’s methodology as well as how we identified commonalities in findings. We employed a comparative thematic synthesis to evaluate the four studies and determine the common findings.
Comment 9: Are figures 1 and 2 pre-existing figures or are they your figures?
Response: These figures are pre-existing from the four studies we are discussing and that have been published. We have added citations to the figure titles from the studies they came from and footnotes under each figure to clarify this.
Comment 10: Section 5 - the references here seem like the review was not necessarily systematic - rather, there was some cherry-picking in terms of the studies. What was being looked for and found, versus what the studies as a group were suggesting? Again, more needs to be done in terms of clarifying the method and explaining what occurred with that.
Response: We thank the reviewer, but we focused on common themes from our studies rather than a wide review of the literature; we wanted a focused message and narrative. This is a broad topic with lots of views, but there are common findings that we have reported here. We have explained that in our methods.
Comment 11: Section 6 - these barriers - there are no references back to studies. What was the rationale with this?  How did you discern the barriers from the literature?
Response: This section discusses the barriers discussed across all four studies, conducted by the authors of this manuscript. These were determined through a synthesis of the findings in the four studies described and through the key informant interviews and focus groups that were conducted in each of the studies. These were recent studies that were based on a wide literature review to include all valid and relevant topics and to be comprehensive and up to date at the time. We acknowledge that hesitancy and attitude towards vaccine are changing rapidly due to politics and economy.
Comment 12: Table 1 - how were these identified/defined as 'key'?
Response: We thank the reviewer for the question and the opportunity to clarify. This table is now Table 2. In section 2.1 we explain that each study included a comprehensive literature review. Additionally, we included more details in the methodology section on our qualitative analysis and how we identified key themes in all four studies. Additionally, in section 3.1, we discuss the comparative thematic synthesis completed to determine the overlapping patterns and “key” findings to share with readers.
Comment 13: Section 7 - it is difficult for me to discern the quality of the discussion, given my lack of clarity on the method and presentation of analysis/findings. I would suggest though that 'building trust' might be different in the eyes of the beholder - and how policy is written leads to strikingly different outcomes?  Also, what the public actually thinks about vaccines as a result of the information/misinformation/sources of information they rely upon.
Response: Thank you for this comment, we hope the changes made throughout the manuscript have provided the context necessary to improve clarity.